# Differential Diagnosis of Hepatic Mass with Central Scar: Focal Nodular Hyperplasia Mimicking Fibrolamellar Hepatocellular Carcinoma

**DOI:** 10.3390/diagnostics12010044

**Published:** 2021-12-27

**Authors:** Teodoro Rudolphi-Solero, Eva María Triviño-Ibáñez, Antonio Medina-Benítez, Javier Fernández-Fernández, Daniel José Rivas-Navas, Alejandro José Pérez-Alonso, Manuel Gómez-Río, Tarik Aroui-Luquin, Antonio Rodríguez-Fernández

**Affiliations:** 1Nuclear Medicine Department, Virgen de las Nieves Hospital, 18014 Granada, Spain; eva_gor@hotmail.com (E.M.T.-I.); javifernandez0210@gmail.com (J.F.-F.); drivas1401@gmail.com (D.J.R.-N.); manuel.gomez.rio.sspa@juntadeandalucia.es (M.G.-R.); arouitariq@gmail.com (T.A.-L.); antonio.rodriguez.fernandez.sspa@juntadeandalucia.es (A.R.-F.); 2Radiology Department, Virgen de las Nieves Hospital, 18014 Granada, Spain; rrabenit@gmail.com; 3Surgery Department, Virgen de las Nieves Hospital, 18014 Granada, Spain; apma85@hotmail.com

**Keywords:** fibrolamellar hepatocellular carcinoma, deshydroxy-(^18^F) fluorocholine, positron emission tomography, focal nodular hyperplasia, central scar hepatic mass

## Abstract

Fibrolamellar hepatocellular carcinoma is a primary hepatic tumor that usually appears in young adults. Radical surgery is considered curative for this kind of tumor, so early diagnosis becomes essential for the prognosis of the patients. The main characteristic of this entity is the central scar, which is the center of differential diagnosis. We report the case of a 30-year-old man who was diagnosed with fibrolamellar hepatocellular carcinoma by ultrasonography. Contrast-enhanced CT confirmed this diagnosis, and the patient underwent a [^18^F] fluorocholine PET/CT. Hypermetabolism and the morphology in the nuclear medicine exploration suggest neoplastic nature of the lesion. Radical surgery was performed, and histopathologic analysis was performed, which resulted in focal nodular hyperplasia. Hepatic masses with central scar could have a difficult differential diagnosis, and focal nodular hyperplasia could mimic fibrolamellar hepatocellular carcinoma imaging patterns. These morphofunctional characteristics have not been described in [^18^F] Fluorocholine PET/CT, so there is a need to find out the potential role PET/CT in the differential diagnosis of hepatic mass with central scar.

## 1. Introduction

Fibrolamellar hepatocellular carcinoma (FLHCC) is a rare primary hepatic tumor that is more frequent in non-cirrhotic young adult populations (5 to 35 years old) [1]. As FLHCC has a mortality rate similar to conventional hepatocellular carcinoma, performing surgery with complete resection is the treatment of choice [2], and it is considered one of the only curative options, so its early diagnosis may affect the prognosis of these patients [3].

From the histopathological point of view, FLHCC is composed of large polygonal cells with big nuclei, marginalized chromatin, and distinguished nucleoli. Cytoplasm usually contains pale bodies and hyaline globules, surrounded by lamellar stroma, composed of collagen deposited in parallel bands. This tumor diagnosis usually needs a confirmatory test, as conventional hepatocellular carcinomas have overlapping features with FLHCC [4].

Ultrasonography (US) is usually the first imaging study that helps in the diagnosis of FLHCC. It is usually described as a solitary, lobulated mass with variable echographic texture. It is characterized by a central scar that could appear as a central area of hyperechogenicity. The next diagnostic step is frequently the computerized tomography (CT) study, in which FLHCC is shown as a lobulated lesion with good edge definition and a central scar with calcification nodules [5]. The enhancement is heterogeneous due to different portions of cellular and fibrous tissue. Delayed images could show an early washout of the vascular area with delayed enhancement of the fibrous lamellae. The absence of late enhancement in the central scar could be useful to distinguish FLHCC from cholangiocarcinoma or focal nodular hyperplasia (FNH) [6].

Nuclear medicine procedures have been described as a useful tool in the diagnosis of FLHCC. Classically, [^99m^Tc]Tc-labelled red blood cells have been used, as it showed increased uptake at arterial phase and washout on delayed phase [7]. The role of the [^18^F]FDG positron emission tomography (PET)/CT is limited in the diagnosis of FLHCC due to increased uptake that only appears in half of the patients [8]. However, radioisotope-labelled choline PET/CT demonstrated more sensitivity than [^18^F]FDG PET/CT in differentiated HCC [9,10,11].

## 2. Case Report

Herein, we report the case of a 30-year-old male who was referred to the gastroenterologist due to an asthenia, low fever (37.7 °C), and transaminase elevation and hepatomegaly as clinical picture. The patient developed a burning and soft pain sensation in the right hypochondrium, more intense after making a significant effort. SARS-CoV-2 infection was dismissed. The patient did not have a significant medical history, other than tonsillectomy and chronic blepharitis.

A standard US was performed, and a solid hepatic mass with lobed contours and hypoechoic capsule were described. This mass was located in segment IV and compressed and displaced the left portal vein and the anterior branch of the right portal vein. It presented arterial vascularization inside and thick calcifications of linear morphology. There were no other significant findings in the study. US concluded that this mass had imaging characteristics that suggested a FLHCC as the first diagnostic option and a multiphase liver CT.

A contrast-enhanced thoracic and abdominopelvic CT with arterial, portal, and late phase were performed. A space-occupying lesion with a central scar and gross calcifications was described. After the intravenous contrast administration, the lesion presented hyper-uptake in the arterial phase, with a portal phase wash-out and a thin hyper-uptaking capsule in the late phase (Figure 1). The vascular compression was similar to ultrasonography. The rest of the study was bland. The findings in the CT also suggested a fibrolamellar hepatocellular carcinoma as the first diagnostic option.

The patient was presented to a multidisciplinary committee and was proposed for excision surgery. For a proper surgical planning, [^18^F]Fluorocholine PET/CT was requested. The study showed a hypermetabolic mass with hypometabolic center in the right hypochondrium that corresponded to a hypodense liver mass in segment IV (Figure 2). In the SUL (SUV—Standardized Uptake Value—normalized by lean body mass) quantification process, the SULmax was 17.19 SUV-lbm (normal hepatic tissue SULmax: 15.15 SUV-lbm), the SULpeak was 14.08 SUV-lbm (normal hepatic tissue SULpeak: 13.73 SUV-lbm), and the SULmean was 10.53 SUV-lbm (normal hepatic tissue SULmean: 7.45 SUV-lbm). The MTV (metabolic tumor volume) was 161.98 cm^3^, and the TLG (total lesion glycolysis) was 1695.81 SUV-lbm.cm^3^. The [^18^F]Fluorocholine PET/CT concluded that this mass was suggestive of neoplastic metabolic activity.

The described tumor was surgically removed without any complications, and the surgical piece was sent to the pathology department for a histopathological exam (Figure 3). The left hepatectomy piece was included in formalin. The serial section showed a tumoral lesion of irregularly nodular subcapsular tumor that measured 7.2 × 7.5 × 6 cm of maximum diameter. The main portion of the tumor had a yellowish appearance with an elastic consistency whitish center of 1.6 × 0.5 cm of maximum diameter. The tumor was in the segment IVb and had less than 0.1 cm of margin and less than 0.1 cm from the liver surface. Resection margins were free of lesions. The pathologist concluded absence of malignant neoplastic tissue, ruling out a fibrolamellar hepatocellular carcinoma. The tumor was identified as a focal nodular hyperplasia secondary to a pre-existing central arterial malformation, with strong evidence in the Masson trichrome and caldesmon stains.

## 3. Discussion

FNH is one of the most common benign hepatic lesions, which usually appears in young and middle-aged women (80% female prevalence). Histologically, FNH is characterized by functioning hepatocytes packed densely with a central scar of fibrous tissue without calcifications [12].

Differential diagnosis of solitary hepatic nodules is sometimes very difficult or even impossible [13]. The mimicry between FNH and FLHCC has been described by some authors, as they can look similar in imaging studies [14]. As FLHCC is a rare tumor with a young age of emergence, in this case, it was necessary to establish it as a possibility. In imaging studies, the US usually shows a homogeneous isoechoic image with a centrifugal arterial flow on Doppler. US is the most commonly used diagnostic study for focal liver lesions first diagnosis; however, contrast-enhanced US have demonstrated a higher usefulness for differential diagnosis. It has been described that, on contrast-enhanced US, the FLHCC shows hyperenhancement in arterial phase and hypoenhancement in portal venous phase and late phases, while FNH usually shows hyperenhancement in portal venous and late phases [15]. In this clinical case, contrast-enhanced US was not contemplated, because standard US and contrast-enhanced CT were clear. In reference to differential diagnosis of these injuries in CT studies, FLHCC is usually described as hypoattenuating lesions in pre-contrast study, heterogeneous hypervascular enhancement in arterial phase, and irregular hyperattenuating or isoattenuating in the portal venous and late phases, while the central scar could show hypoenhancement and calcifications in all phases [16]. Regarding FNH, CT usually shows a hypo or isointense mass surrounded by normal liver tissue. It also shows a contrast enhancement in arterial phase with a non-enhanced scar that is isointense in portal venous and late phases [17]. Regarding MR, FLHCC is usually hypointense in T1-weighted and hyperintense in T2-weighted MRI, with a central scar in both sequences, but inner calcifications could be difficult to see. In some studies, authors reported that FLHCC does not enhance with hepatobiliary-specific contrast [7]. FNH on MR is described as an isointense or hypointense mass in T1-weighted MRI and hyperintense or isointense in T2-weighted MRI, with a central scar with a homogenous enhancement in arterial phase and late-phase enhancement with hepatobiliary gadolinium [17]. In this case, MR was not performed, because the radiologist was clearly convinced of the FLHCC diagnosis due to the calcification’s presence in the scar, the portal phase wash-out of the lesion, and the presence of capsule, which are very uncommon findings in FNH. Following ESMO clinical practice guidelines, contrast-enhanced CT could be valid [18], although MR with extracellular contrast have a higher sensitivity, only significant in small nodules, but similar specificity in all lesions [19].

On several occasions, similar radiologic patterns between hepatocellular carcinoma (HCC with central scar) and FNH have been described [20], but these analogous images have not been described in [^18^F]Fluorocholine PET/CT. In comparison with CT and MR, radioisotope-labelled choline PET/CT have demonstrated higher sensitivity and specificity at hepatic lesion diagnosis and higher accuracy in extrahepatic lesions diagnosis [21] and could alter the management of a third of the patients at least [22]. It was pointed out that radioisotope-labelled choline PET/CT could be useful to differentiated FNH from other benign lesion [23,24], but FNH is usually seen as a focus in the liver parenchyma in the majority of cases, so it could be a source of false-positive result, as in this case [25], or as a metastasis in other neoplastic studies, such as a prostate cancer study [26]. Additionally, some authors raised the possibility that central elements, such as scars or necrosis, could hinder the characterization of liver masses [27], inner calcification being the main differential characteristic [28]. However, this pathology has not been well studied, as it is frequently excluded from clinical trials [20], so there is a need for more studies that allow for better understanding of this entity [29,30].

The main limitation of this case could be the lack of MR images to support the diagnosis. However, MR patterns between FNH and FLHCC could be close, so there are doubts about the real need of this exploration [27]. Additionally, the lack of literature about the role of radioisotope-labelled choline PET/CT could obstruct the assessment of the finding in this study.

## 4. Conclusions

The correct characterization of liver lesions could be difficult in some situations, which can make the patient follow the wrong direction in the clinical algorithm. In this case, ultrasonography, contrast-enhanced CT, and [^18^F] Fluorocholine PET/CT oriented the diagnosis to a fibrolamellar hepatocellular carcinoma. The patient underwent an unnecessary surgery, and a diagnosis of a focal nodular hyperplasia was concluded. Further studies could establish useful criteria to discern between these similar liver tumors to improve health care and decrease iatrogenic complications.

## Figures and Tables

**Figure 1 diagnostics-12-00044-f001:**
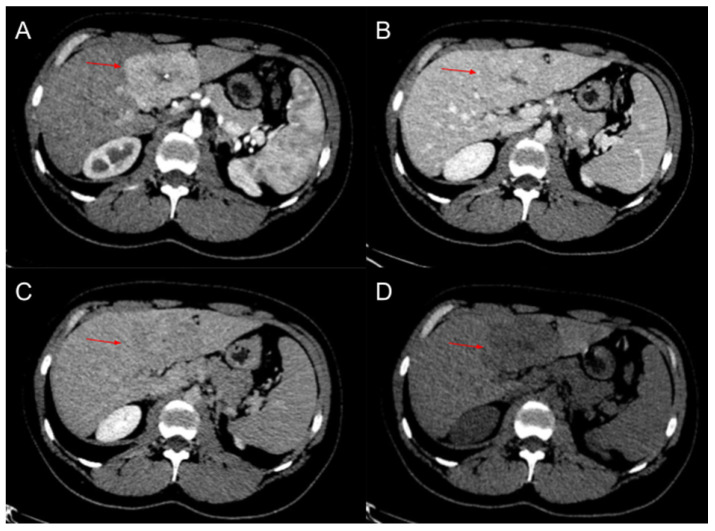
Abdominopelvic contrast-enhanced CT with a space-occupying lesion (arrow) with a central scar. (**A**): Arterial phase. (**B**): Portal phase. (**C**): Late phase. (**D**): Non-contrast-enhanced CT.

**Figure 2 diagnostics-12-00044-f002:**
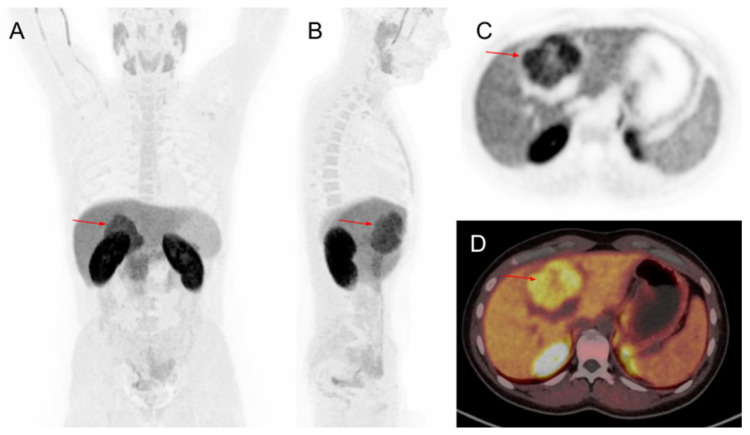
A hypermetabolic mass (arrow) with hypometabolic center in a [^18^F]Fluorocholine PET/CT. (**A**): MIP frontal view. (**B**): MIP side view. (**C**): Abdomen axial slice. (**D**): Abdomen PET/CT fusion axial slice.

**Figure 3 diagnostics-12-00044-f003:**
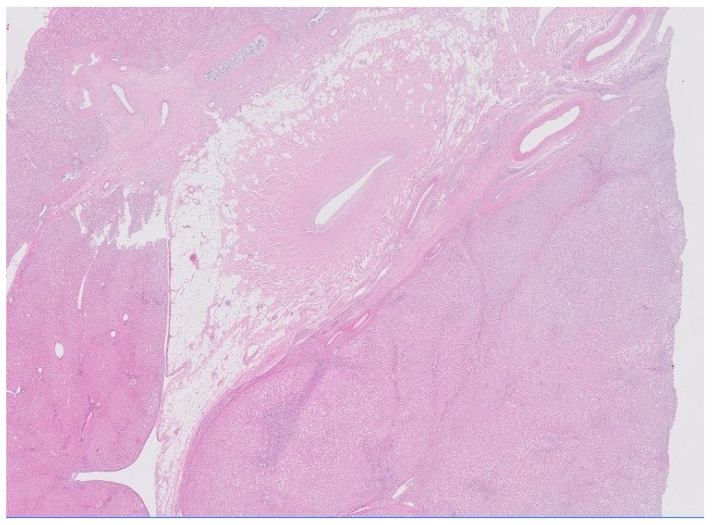
Histologic 4x image. Macroscopically, a solitary non-encapsulated lesion with nodular appearance and subcapsular localization was identified. Its measures were 7.2 × 5.7 × 6 cm, with yellowish coloration and a whitish central area. Histologically, it was composed of hepatocytes without atypia (absence of nuclear pleomorphism and mitotic figures) and surrounded by fibrous septa with arterial vessels and variable degree of reactive bile ducts. In the periphery of the lesion, a large vessel with a thick muscular wall surrounded by fibrous band was observed in relation to a pre-existing arterial malformation.

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
