# Peer review of "Differential Diagnosis of Hepatic Mass with Central Scar: Focal Nodular Hyperplasia Mimicking Fibrolamellar Hepatocellular Carcinoma"

_diagnostics, 2021, doi:10.3390/diagnostics12010044_

Round 1
Reviewer 1 Report
This is an interesting case report on FNH differential diagnosis.
The paper is worth publishing after some (minor) revision. However, the main limitation of this study is the lack of MR images.
I propose to add a comment in the discussion about contrast-enhanced ultrasonography.
I also propose to add pictures from histopathologic examinations of liver mass.
Author Response
Reviewer 1 answer
This is an interesting case report on FNH differential diagnosis.
The paper is worth publishing after some (minor) revision. However, the main limitation of this study is the lack of MR images.
Thank you so much for your words. We appreciate your comments.
I propose to add a comment in the discussion about contrast-enhanced ultrasonography.
Thank you for this point. We have included a comment about as it has been recommended.
I also propose to add pictures from histopathologic examinations of liver mass
Thank you for your comment. We have added this picture with its caption.
Reviewer 2 Report
The authors present the case of a young adult with a liver tumor that was considered fibrolamellar hepatocarcinoma, and after surgery, the histology revealed the focal nodular hyperplasia. The manuscript emphasizes the need for differential diagnosis in such a liver tumor.
This case report is, in general, well written, presenting the most critical aspects of the diagnosis and management of the case, including a discussion on the differential diagnosis.
I suggest including a paragraph in the Discussion with a more precise comparison between focal nodular hyperplasia and fibrolamellar hepatocarcinoma, including all relevant aspects. Imaging comparations may be presented more in-depth for this differential diagnosis.
The authors should correct the manuscript for the English in some sentences (ex: line 30 "than", line 57-58 gastroenterologist instead of digestive system specialist, line 60 SARS-CoV-2, do not use haven't in scientific text), but also for better phrases (as in lines 29-33). The abbreviated words should be used in the manuscript after their definition at the first appearance.
Author Response
Reviewer 2 answer
The authors present the case of a young adult with a liver tumor that was considered fibrolamellar hepatocarcinoma, and after surgery, the histology revealed the focal nodular hyperplasia. The manuscript emphasizes the need for differential diagnosis in such a liver tumor.
This case report is, in general, well written, presenting the most critical aspects of the diagnosis and management of the case, including a discussion on the differential diagnosis.
Thank you so much for your comments. We believe that this clinical case could be interesting to draw attention about the importance of hepatic mass differential diagnosis.
I suggest including a paragraph in the Discussion with a more precise comparison between focal nodular hyperplasia and fibrolamellar hepatocarcinoma, including all relevant aspects. Imaging comparations may be presented more in-depth for this differential diagnosis.
We appreciate you recommendation and we have added a intense comparation between imaging techniques in the study of this two entities.
The authors should correct the manuscript for the English in some sentences (ex: line 30 "than", line 57-58 gastroenterologist instead of digestive system specialist, line 60 SARS-CoV-2, do not use haven't in scientific text), but also for better phrases (as in lines 29-33). The abbreviated words should be used in the manuscript after their definition at the first appearance.
Thank you for your comment. Mistakes have been corrected as requested and some sentences have been reformulated.